# Mpox-Related Stigma Among Gay, Bisexual, and Other Men Who Have Sex with Men: A Narrative Review

**DOI:** 10.3390/healthcare13212690

**Published:** 2025-10-23

**Authors:** Matthew N. Berger, Chenoa Cassidy-Matthews, Marian W. A. Farag, Cristyn Davies, Rohan I. Bopage, Shailendra Sawleshwarkar

**Affiliations:** 1Specialty of Child and Adolescent Health, Faculty of Medicine and Health, The University of Sydney, Westmead, NSW 2145, Australia; cristyn.davies@sydney.edu.au; 2Sydney Infectious Diseases Institute, Faculty of Medicine and Health, The University of Sydney, Camperdown, NSW 2050, Australia; 3Office of the Chief Medical Health Officer, Vancouver Coastal Health, Vancouver, BC V5Z 4C2, Canada; 4Vancouver Coastal Health Research Institute, Vancouver Coastal Health, Vancouver, BC V5Z 1M9, Canada; 5Hillarys Plaza Medical Centre, Hillarys, WA 6025, Australia; marian.farag@hillarysplazamc.com; 6School of Social Sciences, Western Sydney University, Locked Bag 1797, Penrith, NSW 2751, Australia; 7Westmead Clinical School, Faculty of Medicine and Health, The University of Sydney, Westmead, NSW 2145, Australia; 8Western Sydney Sexual Health Centre, Western Sydney Local Health District, Parramatta, NSW 2150, Australia; 9Sydney Medical School, Faculty of Medicine and Health, The University of Sydney, Camperdown, NSW 2050, Australia

**Keywords:** mpox, stigma, GBMSM, LGBTQ, public health

## Abstract

**Introduction:** Mpox emerged as a multi-country outbreak in 2022 and disproportionately affected gay, bisexual, and other men who have sex with men (GBMSM). Stigma is known to exacerbate health crises by discouraging testing, treatment, and vaccination. This review aimed to explore stigma associated with Mpox among GBMSM from July 2022, when mpox was declared a public health emergency of international concern. **Methods:** The PICO framework guided this narrative review. A search was conducted across the following databases from inception to June 2025: PubMed/MEDLINE, Embase, CINAHL, and Web of Science. The literature had to be empirical, peer-reviewed research that focused on mpox-related stigma in GBMSM. **Results:** Forty-seven studies were included in this review. The following themes were derived: (1) healthcare experiences, (2) media influence, (3) internalised and anticipated stigma, (4) public health messaging, (5) community responses, and (6) psychosocial impact. Healthcare experiences were marked by anticipated discrimination; many GBMSM delayed testing or vaccination for fear of being disclosed or labelled promiscuous. This was especially apparent in contexts where same-sex relationships are criminalised, leading some men to self-medicate or seek clandestine services. Media analyses revealed that social and traditional platforms often amplified blame and homophobia, though community-led counter-messaging helped shift narratives. Internalised and anticipated stigma resulted in shame, concealment of symptoms, avoidance of care, and heightened anxiety. Public health messaging that framed mpox as a behaviour-linked rather than identity-linked risk was more acceptable, and flexible vaccination strategies (e.g., offering less conspicuous injection sites) increased uptake. Stigma contributed to psychosocial distress and may have impeded outbreak control. **Conclusions:** Mpox-related stigma among GBMSM operates at individual, community, and structural levels, echoing patterns from the HIV era. Effective mitigation requires rights-based, destigmatising communication, culturally competent care, and collaboration. Addressing stigma is vital to controlling future outbreaks and ensuring equitable healthcare access.

## 1. Introduction

The mpox (formerly known as monkeypox) outbreak in 2022, which was declared a public health emergency of international concern (PHEIC) by the World Health Organization, disproportionately affected sexually minoritised cis and trans men, including gay, bisexual, and other men who have sex with men (GBMSM) [1]. The outbreak was primarily attributed to the monkeypox virus (MPXV) clade IIb (lineage B.1 and A.2) [2]. Clade I primarily circulates in Africa but has been reported in countries outside of Africa as travel-associated infections [3]. MPXV is an orthopoxvirus that was first isolated in 1958 from cynomolgus monkeys, with the first human case reported in 1970 [4]. The natural reservoir of MPXV has not yet been determined; however, it has been isolated from several African rodents [5]. Mpox is typically self-limiting, with signs and symptoms lasting two to four weeks. Signs and symptoms can include lymphadenopathy, fever (≥38 °C) or history of fever, headache, myalgia, arthralgia, back pain, and sore throat. Prodromal symptoms may not be present in all cases. A maculopapular rash and skin lesions are common; lesions take between 14 and 21 days to scab over and resolve [6,7,8]. Case fatality rates are estimated to be 3–11% for clade Ia, <1% for clade Ib, 1% for clade IIa, and <1% for clade IIb [9]. Case fatality rates may be associated with lower healthcare accessibility and resources in countries affected by mpox and thus may be lower in high-income countries [10].

Research examining mpox among GBMSM found significant associations with unprotected or condomless receptive anal sex, contact with mpox cases, multiple sexual partners, and coexisting HIV or STIs [11,12]. In response, early in the mpox outbreak, GBMSM adopted various strategies to reduce risk, including reducing the number of sexual partners [13]. Those at greater risk of severe disease include people who are unvaccinated, immunocompromised (particularly those with poorly controlled human immunodeficiency virus (HIV)), pregnant, and children [10,12,14,15]. A safe and effective vaccine against Mpox is available and has demonstrated its effectiveness in preventing mpox disease. Two doses are recommended, but both one and two doses have demonstrated protection [16]. Mpox severity and duration are considerably reduced in those with previous immunity through immunisation or previous infection [17]. In many high-income countries, mpox vaccination is recommended and publicly funded for people at a higher risk of contracting mpox, including GBMSM, transgender people, and contacts of cases [18,19]. These public health efforts, while important, must be understood within broader sociocultural contexts where stigma continues to shape both public perceptions of mpox and the lived experiences of those most affected. Monkeypox was renamed to mpox in 2022 to reduce the stigma and racism associated with its name [20].

Stigma refers to the combined processes of labelling, stereotyping, separation, status loss, and discrimination, all occurring within a power context. Broader than racism or discrimination alone, it applies to various characteristics, such as sexual orientation, disability, HIV status, or obesity, and includes, but is not limited to, unequal treatment [21,22]. Many countries still criminalise same-sex relationships, with a handful punishing same-sex acts by death [23,24]. Lesbian, gay, bisexual, transgender, queer and questioning, plus [representing other sexual orientations and gender identities not included in the acronym] (LGBTQ+) people tend to utilise digital platforms, particularly social media, to explore identity, connect with likeminded people, and seek social support to avoid the potential consequences of outing themselves to their offline networks (i.e., family and friends) [25]. Stigma and discrimination remain an issue for GBMSM even in countries that have decriminalised same-sex relationships [22]. This is evident even in healthcare, which is expected to provide safe and inclusive spaces, but often fails to do so because of structural barriers (e.g., policies or institutional practices) and interpersonal barriers (e.g., provider bias or discriminatory behaviour). This creates issues where GBMSM feel they need to avoid or delay accessing healthcare. Healthcare professionals tend to lack appropriate, culturally safe practices tailored for LGBTQ+ individuals, with gaps evident both explicitly (such as discriminatory remarks and refusal of care) and implicitly (such as heteronormative assumptions and lack of inclusive language) [22,26].

This review focusses on the levels of stigma and its impact on individual, community, and structural levels among GBMSM since the mpox PHEIC declaration. This narrative review aims to synthesise empirical research on mpox-related stigma among GBMSM since the 2022 global outbreak. It explores the types and contexts of stigma experienced, its impact on health behaviours and mental well-being, and the influence of media and public health messaging. This review also highlights community and institutional responses to mitigate stigma and identifies key gaps in the literature to inform future research and public health strategies.

## 2. Methods

This narrative review sought to explore the current literature on mpox-related stigma using a PICO [Patient/Population, Intervention, Comparator/Control, and Outcome(s)] framework approach. To be included in this review, studies were required to be (1) empirical in nature; (2) focused on stigma; (3) related to mpox; (4) involve GBMSM, (5) peer-reviewed, and (6) containing data that was collected after July 2022 (or published in or after 2022 if study dates were unavailable). Given the scope of this topic, no restriction was applied to the age of the population or the need for a comparator. Additionally, no restriction to language was applied. We searched the following databases: MEDLINE, Embase, CINAHL, and Web of Science from inception to June 2025. Google Scholar and manual searching of reference lists in the included studies was performed to capture other relevant records. The below search strategy was used (search strategies are further detailed in Appendix A):

TS = (“mpox” OR “monkeypox” OR “monkey pox” OR “MPXV”) AND TS = (“stigma” OR “stigmati*ation” OR “discrimination” OR “social perception” OR “social exclusion” OR “internali*ed stigma”) AND TS = (“men who have sex with men” OR “MSM” OR “gay” OR “bisexual” OR “GBMSM” OR “sexual minority men” OR “queer men”)

Title and abstract screening were followed by full-text review for potentially eligible studies. Inclusion decisions were made based on relevance to the review aims and methodological clarity. Title and abstract screening and full-text reviews were completed by MNB and MWAF using Covidence (Veritas Health Innovation, Melbourne, Australia) [27]. Data extracted from included studies included the following: study setting, design, participant characteristics, type of stigma examined, key findings, and implications. This phase was carried out by authors MNB and CCM. A narrative synthesis was conducted to summarise patterns and insights across diverse study designs and settings. This review was not registered in PROSPERO and did not follow PRISMA guidelines [28] strictly, as it was intended as a narrative rather than systematic review; however, transparent reporting and structured synthesis were prioritised throughout the process.

## 3. Results

Forty-seven studies were included that focussed on mpox-related stigma among GBMSM. Figure 1 illustrates the screening and inclusion process. Included studies were based mostly in the United States (US) (n = 18, 28.6%), China (n = 6, 9.5%), the United Kingdom (UK) (n = 6, 9.5%), Brazil (n = 5, 7.94%), Spain (n = 4, 6.4), Australia (n = 3, 4.8), and Canada (n = 3, 4.8). The majority were cross-sectional studies (N = 25, 43.1%), followed by qualitative studies (n = 21, 36.2%), and content/media analysis studies (n = 5, 8.6%). Further information on these studies is detailed in Table 1.

### 3.1. Healthcare Experiences

Since mpox was declared a PHEIC in 2022, GBMSM have carried a disproportionate burden of both disease and stigma. A qualitative study of mpox cases in the US revealed that despite limited mpox vaccine supply, GBMSM were motivated to receive immunisation, often queuing at LGBTQ+ community sites (e.g., bathhouses or events) [46]. This study indicated that community-based strategies, coupled with existing healthcare relationships, can foster rapid vaccine uptake among GBMSM [46]. There were further positive experiences; clinics in England offered compassionate case management and active follow-up, which participants said helped them adhere to isolation and recover without feeling marginalised [70]. However, some participants feared that receiving vaccination would visibly label them as gay because the intradermal forearm injection left a tell-tale scar [29]. These findings echo earlier observations by Bergman and colleagues (2022), who reported that MSM and people with HIV experienced stigma from family, healthcare providers, and broader communities, and called for nursing interventions to counteract such discrimination [30].

In the United Kingdom, community-led initiatives bridged trust gaps between sexual health clinics and local LGBTQ+ populations; these efforts ensured high demand for vaccines and minimised the effect of stigma, but the authors criticised national public health authorities for under-resourcing the response and communicating poorly [31]. Brazilian research highlighted structural vulnerabilities: a Portuguese language study described how GBMSM faced barriers, including employment precarity, racial discrimination, and lack of access to culturally competent care [32]. Another Brazilian study documented self-care practices adopted during the outbreak: many men intensified hygiene routines, limited physical contact, inspected skin lesions, used masks, practised safer sex, and challenged misinformation; the authors emphasised nurses’ roles in guiding these practices and confronting stigma [33].

In France, cisgender GBMSM diagnosed with mpox were not surprised to test positive (given high exposure and targeted campaigns) but were unprepared for the severity of their symptoms and the degree of social isolation they encountered [38]. These participants described being treated like “plague victims,” encountering erratic care pathways and negotiating with health providers, partners, and employers to maintain privacy and continue working [38]. Qualitative interviews with Chicago GBMSM similarly revealed confusion about where to seek care; men compared the mpox outbreak with previous crises such as HIV and COVID-19 and noted that layered stigmas and fatigue impeded health seeking [46]. Several studies suggested that healthcare experiences differ by race and location [38,40,50,65]. Nigerian GBMSM avoided public clinics altogether because same-sex relationships are criminalised and medical settings felt unsafe; instead, they self-medicated or relied on underground “key population friendly” services [48]. In the US, Black GBMSM receiving mpox vaccinations reported lower trust in health institutions but high willingness to vaccinate when outreach occurred through trusted community organisations [58]. Chinese patients described variable quality of care, and some felt abandoned by providers after diagnosis [73].

### 3.2. Media Influence

Public discourse played a major role in shaping stigma. Social media analysis demonstrates how quickly negative stereotypes proliferated: a multilingual sentiment study of 1.3 million posts across X (formerly known as Twitter) and Facebook found that mpox discussions were highly intertwined with LGBTQ-related terms and that negative sentiments overwhelmingly dominated [56]. A separate content analysis of posts on X (tweets) revealed that users often compared mpox to HIV, recycled conspiracy theories about punishment for sexual behaviour, and spread misinformation that fuelled discrimination and sometimes violence [39,42,56]. Thematic analysis on Reddit posts documented similar patterns: members of key populations shared fears of being outed and ridiculed online, while others pushed back with humour and fact-checking [45]. Scholars argued that mainstream news reproduced tropes from the 1980s by linking the outbreak to gay male sexuality and evoking moral panic [43]. Community-driven messaging offered a counter-narrative; LGBTQ+ organisations launched campaigns and collaborated with journalists to centre human stories rather than sensationalism [40,41,57]. Studies assessing mpox media consumption among primarily GBMSM found that audiences preferred detailed, non-judgmental information delivered through trusted channels such as LGBTQ+ newsletters or dedicated health websites and were less likely to trust brief mainstream news reports [60]. Despite these efforts, the infodemic (excessive spread of false or misleading information during a disease outbreak) illustrated the need for proactive risk communication strategies that anticipate stigmatising frames and provide clear, empathetic messaging from the outset [39].

### 3.3. Internalised and Anticipated Stigma

As public narratives circulated, many GBMSM internalised stigma and anticipated discrimination. In a Spanish cross-sectional study, one-third of MSM avoided seeking medical care for mpox symptoms because they expected judgement from healthcare professionals [53]. Qualitative studies in the US, Canada, and Singapore showed that men hid lesions, downplayed symptoms, and delayed testing to avoid being perceived as promiscuous or “dirty,” sometimes even concealing their sexual orientation from contact tracers [34,35,50]. Fear of visible vaccine scars discouraged vaccination; one cross-sectional survey found that concerns about cosmetic appearance and others’ views strongly influenced preference for receiving the mpox vaccine in less noticeable sites rather than the forearm [65]. Identity concealment had downstream effects: participants who were not open about their sexuality were significantly less likely to learn about vaccination campaigns, to access testing, or to adhere to isolation guidelines [51]. Several studies underscored how internalised stigma intersects with racism, HIV status, and economic precarity. In a US survey, Black and Latino sexual minority men expressed concerns that the outbreak could be exploited to reinforce negative stereotypes about gay men and restrict access to HIV pre-exposure prophylaxis [44]. Young GBMSM in the US Midwest indicated that mpox was perceived as a continuation of past associations between infectious diseases and gay communities, reflecting the persistence of stigma and trauma across generations [67]. In the Netherlands, researchers compared stigma related to mpox with stigma surrounding HIV, syphilis, and COVID-19. They found that mpox was perceived as less deadly but more shameful because of its association with sexual behaviour [75].

Fear of stigma also shaped mental health. Across studies from China, Spain, and Brazil, high mpox risk perception correlated with elevated anxiety, depressive symptoms, and sleep disturbance [52,53,68]. A survey of Chinese GBMSM who experienced both COVID-19 lockdowns and the mpox outbreak reported multifaceted stressors and noted that dual threats magnified psychological distress [52]. Participants in France and Canada described shame and isolation, with some refusing to tell housemates or employers about their diagnosis to avoid eviction or workplace discrimination. At the same time, some men internalised public health messaging to the point of self-blame: they reduced sexual activity not only to prevent infection but to avoid being judged by peers as irresponsible. Respondents emphasised the need for counselling services and peer support groups to help cope with internalised stigma and fear of social rejection [53,54].

### 3.4. Public Health Messaging

Communicating risk without perpetuating stigma proved challenging. In South Korea, a randomised survey experiment demonstrated that describing mpox as concentrated among GBMSM increased support for punitive policies such as banning pride events or mandating segregation [37]. This effect underscores how highlighting a specific identity can activate prejudice, even when epidemiologically accurate. To navigate this, health agencies gradually shifted from identity-based warnings to behaviour-based advice (e.g., multiple close contacts increase risk) and paired targeted outreach with anti-discrimination language. In Singapore, cross-sectional surveys combining quantitative and qualitative data found high vaccine receptiveness among GBMSM when messages emphasised scientific facts, solidarity, and inclusivity, and when officials engaged directly with community leaders [35]. Studies of UK and Irish public health campaigns argued that early messaging was too generic and delayed, forcing gay men to rely on their networks for information; once authorities collaborated with community groups, uptake of vaccines and adherence to isolation improved [31,55].

Researchers also highlighted the importance of addressing vaccine hesitancy and structural barriers; surveys in Brazil revealed that although knowledge and willingness to vaccinate were high (96.9% and 95.1%, respectively), 84.4% of respondents believed that LGBTQIA+ [lesbian, gay, bisexual, transgender, queer, intersex, asexual, plus] people were discriminated against and stigmatised due to mpox. Those diagnosed with mpox reported more sexual partners, substance use (including binge drinking and chemsex), and sex venue attendance and were more likely to change their sexual behaviour after the outbreak, likely driven by a greater awareness and fear of mpox [68]. Another Brazilian study found that chemsex practice (defined as the intentional consumption of psychoactive substances prior to or during sexual activity) during the outbreak was associated with both elevated risk and heightened stigma, suggesting that harm reduction messaging must acknowledge sexual subcultures rather than condemn them [62]. Community-led pop-up vaccination clinics in gay bars and pride events allowed people to receive vaccinations discreetly and to ask questions without fear; these programmes were cited as models for rights-based emergency responses [31,41]. In Taiwan, offering participants a choice of injection site (deltoid vs. forearm) increased vaccine acceptance among those concerned about visible marks [65]. To address underrepresented groups, researchers called for multilingual messaging, collaboration with dating apps, and the inclusion of trans and nonbinary voices in campaign design [55,68]. Many authors stressed that interventions should anticipate intersectional stigma and not rely solely on sexual health clinics; primary care physicians, mental health providers, and community health workers all require training in culturally competent, anti-stigma communication [30,49].

### 3.5. Community Responses

LGBTQ+ communities played a central role in combating mpox stigma and disseminating accurate information. Social networks were not only channels for misinformation and stigma but also powerful tools for resilience. A qualitative study of European activists described how networks of volunteers rapidly produced multilingual fact sheets, fact-checked rumours, and shared lived experiences to normalise the discussion of symptoms, recovery, and challenging shame [40]. In the UK, community health promotion led by sexual health clinics and LGBTQ+ organisations built trust among priority populations and reduced fears of “outing” [31]. Irish community leaders criticised tokenistic national engagement but praised local clinics that worked closely with community partners and used humour and plain language to engage the public and counter stigma [31]. Studies from the Netherlands and Spain emphasised the importance of online support groups and peer-to-peer counselling; participants said these spaces helped them process shame and encouraged them to seek care [75]. Research in Brazil and Peru documented the crucial role of LGBTQ+ and sex worker organisations in distributing condoms, lubricant, and informational materials alongside mpox vaccines; these efforts recognised that sexual health services must address all facets of well-being, including pleasure and safety [61,62].

Public statements by international associations urged the media to stop using images of black hands with lesions unless reporting on African cases and to avoid implying that only gay men are at risk [56,57]. In some countries, activist lawyers successfully challenged discriminatory public health measures (such as police crackdowns on gay venues), reinforcing the message that stigma is not a legitimate public health tool [37,49]. Despite these successes, community groups faced burnout and limited resources; authors called on governments to fund grassroots organisations and to integrate them into emergency planning rather than relying on unpaid labour [31]. Moreover, many respondents cautioned that community initiatives often failed to reach closeted individuals or racial minorities not connected to mainstream LGBTQ+ networks; targeted strategies for these groups remain a gap [67,68].

### 3.6. Psychosocial Impact

Mpox stigma compounded existing mental health challenges among GBMSM. Surveys and interviews across multiple studies documented heightened anxiety, depression, anger, and loneliness [52,53,59,64,66,67,70,74]. In Spain, more than half of respondents reported low mood or anxiety during the outbreak; those most worried about mpox were especially likely to meet the criteria for depression. Chinese studies found that dual threats from COVID-19 and mpox led to pervasive rumination and sleep disturbance [44]. Many men feared being socially isolated if outed as having mpox, leading some to withdraw from queer spaces or to hide symptoms [38,41]. Interviews in Baltimore (US) revealed that people diagnosed with mpox experienced both physical pain and stigma; the visibility of lesions elicited staring and whispers from strangers, intensifying feelings of shame [64]. For Black and Latino GBMSM, historical experiences of medical racism and HIV stigma intensified stress; participants expressed despair that yet another virus was being portrayed as an infection exclusively affecting GBMSM [44]. At the same time, resilience was evident: respondents described using humour and activism to cope, and many reported that receiving support from peers and seeing community mobilisation alleviated some psychological distress. However, mental health services were often unavailable or not tailored to LGBTQ+ clients; studies highlighted the need for accessible counselling and support groups integrated into mpox care [53,54].

## 4. Discussion

Emerging evidence from the reviewed studies underscores that mpox stigma has shared similar patterns seen in the early HIV pandemic and that stigma adversely affected mental health, well-being, and health behaviours [30,43,52,53,66,67]. GBMSM across different regions feared being labelled as vectors of disease and faced discrimination when seeking care [34,38,46,48,50,51]. GBMSM anticipated that early media and healthcare communication would categorise mpox as a “gay disease” [34,42,43,67,75]. Mpox-related stigma led GBMSM to avoid healthcare services, including vaccination, to avoid accidental disclosure of their sexuality [29,34,50,51,68]. Policy adjustments can mitigate stigma’s impact; several studies found that GBMSM preferred less visible vaccine injection sites (e.g., upper arm or thigh instead of forearm) to avoid identity disclosure by vaccine scars [29,35,62,65]. Similarly, healthcare professionals noted that mpox-related stigma affected GBMSM clients, leading some to avoid health services, fear being outed, or give misleading information during contact tracing [76]. Healthcare professionals need to endorse a non-heteronormative approach when conducting case investigations or providing healthcare [77,78]. By July 2024, more than 102,000 mpox cases and 220 deaths had been reported across 121 countries [79]. The high case counts, combined with the concentration among GBMSM, created fertile ground for stigma and homophobic narratives [67,70,80].

Despite these challenges, community-led initiatives provided effective countermeasures. UK sexual health clinics partnered with LGBTQ organisations to deliver respectful care and inclusive messaging, helping to maintain vaccine uptake and trust [31]. Social media campaigns and peer-education efforts corrected misinformation and normalised recovery stories, highlighting the importance of involving affected communities in public health responses. Collectively, the evidence suggests that stigma operates at individual and structural levels, amplifying fear and secrecy and impeding outbreak control, while community-driven engagement and destigmatising communication can mitigate its harms [31]. In countries where same-sex relationships remain criminalised, some evidence of community-driven engagement was observed. For example, a Nigerian study found that GBMSM often avoided public health clinics and instead relied on self-medication or underground services [48]. The move from Twitter to X has resulted in increased hate speech, including homophobia and transphobia [81,82]. Legislation has emerged in numerous countries, and political platforms, for example, in Poland and the United States, advocate for removing protections for LGBTQ+ people [83,84,85]. Such legal environments and stigmatisation not only embolden perpetrators of hate crimes but also drive marginalised men further from health services, complicating disease surveillance and outbreak control [48,84]. Mpox highlighted the importance of an ethically grounded response that addresses systemic barriers, including stigma, through community engagement and the use of culturally safe and inclusive language that guides medical communication [86,87,88,89]. The Health Stigma and Discrimination Framework is a possible tool that can be implemented to develop outbreak-specific response strategies [90].

The insights from this review point to several actionable strategies to address mpox-related stigma. Public health communication should be crafted to inform without insinuating blame. Public health and media must avoid perpetuating the notion that mpox is an exclusively GBMSM disease, instead emphasising that mpox can affect anyone and focusing on behaviours or contexts of transmission rather than identities [54,75]. Campaigns should involve LGBTQ+ community representatives from inception to dissemination to ensure language is sensitive and inclusive [31]. These partnerships can facilitate accurate information dissemination, support those in isolation, and combat misinformation on social media [31]. Healthcare professionals require tailored, culturally safe training to recognise and reduce stigma in healthcare settings. Several studies noted that MSM may avoid or delay seeking care for mpox due to anticipated judgement by providers, especially those who are also living with HIV or from racial minority groups who face layered stigma [54]. Cultural competency training and assurances of confidentiality can encourage timely care-seeking [26,67]. In sum, an effective stigma-mitigation strategy for mpox requires health communication, healthcare delivery, and community engagement to be aligned in a way that respects and protects the affected communities [31,91,92,93].

### 4.1. Limitations

Several limitations must be acknowledged. As a narrative review, our search strategy may not have captured all relevant studies, and we did not perform a formal quality appraisal of each study. The included studies were relatively few and varied widely, precluding any quantitative meta-analysis of stigma prevalence. The heterogeneity in study populations and measures (from stigma scales to interview themes) means our synthesis is largely descriptive. Many studies had small or localised samples, which may limit generalisability. Most evidence came from high-income countries with active LGBTQ+ health infrastructures; thus, experiences of mpox stigma in other regions were not represented. Additionally, the rapidly evolving context of the 2022 outbreak means that social attitudes could shift over time; the included studies largely captured early-outbreak sentiments. While parallels between HIV and mpox are informative, it is important to acknowledge the limits of this comparison, as the social, epidemiological, and healthcare contexts differ across time and setting. Despite these limitations, this review provides timely insights during an ongoing public health challenge.

### 4.2. Future Directions and Research

Research on mpox-related stigma among GBMSM has expanded rapidly since 2022, but several gaps remain. Most studies are cross-sectional and qualitative and concentrated in high-income countries, limiting our understanding of stigma in low- and middle-income countries where healthcare access and cultural factors differ. Longitudinal research is needed to examine how stigma evolves across different outbreak phases and its lasting mental health impacts. Additionally, few studies have explored intersectional stigma among subgroups, including racial and ethnic minorities and gender-diverse people (i.e., trans men and non-binary people), despite evidence of disproportionate vulnerability [44,58,94]. Stronger integration of behavioural science into public health messaging is needed to address vaccine hesitancy and identity concealment. Addressing these gaps could inform more equitable, stigma-sensitive approaches to outbreak response and preparedness [31,70,86].

## 5. Conclusions

This review underscores that the 2022 mpox outbreak exposed and amplified longstanding structural and interpersonal stigma facing GBMSM. The initial focus on GBMSM as a high-risk group fuelled scapegoating and moralising narratives reminiscent of the early HIV pandemic, leading many men to anticipate or experience stigma in healthcare, delay testing, conceal their identity, or avoid vaccination. However, healthcare services that had strong community ties providing HIV and COVID-19 prevention and management may have increased trust during the mpox outbreak. Stigma was also catalysed by sensationalist media coverage and online misinformation, which portrayed mpox as exclusively affecting GBMSM and sometimes provoked harassment and violence. This review highlights promising strategies: community-led health promotion, culturally safe healthcare services, co-created and behaviour-based messaging, and peer support networks all helped reduce stigma, build trust, and improve vaccine uptake. Inclusive public health campaigns framed mpox as a human issue rather than a moral failing, while activists and healthcare professionals advocated for rights-based approaches. Rising anti-LGBTQ+ sentiment in some settings and countries threatens to undo gains and may intensify stigma if left unchallenged. Future responses must therefore integrate destigmatising communication, community engagement, and mental health support into outbreak planning.

## Figures and Tables

**Figure 1 healthcare-13-02690-f001:**
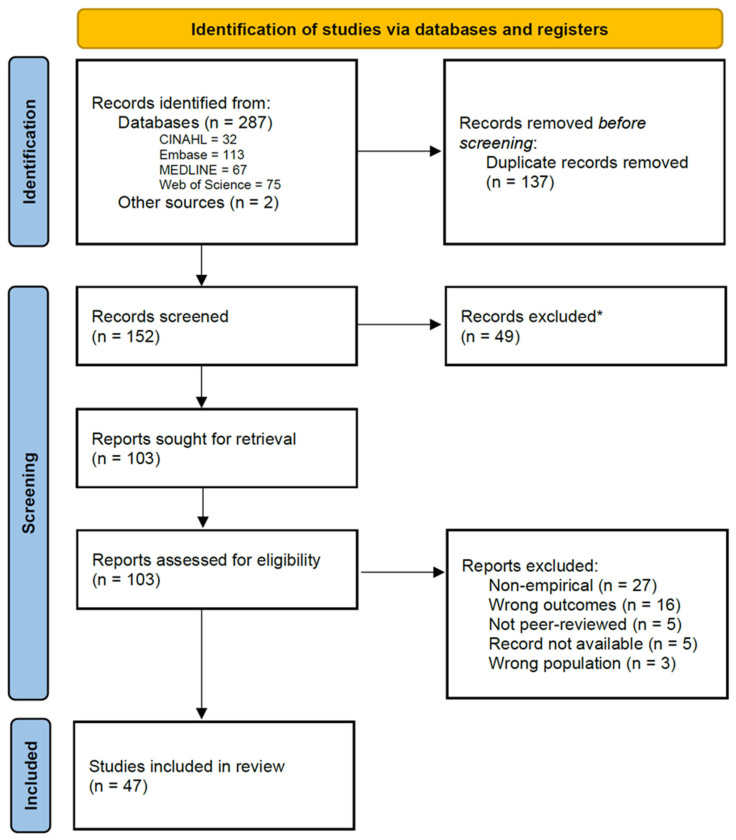
PRISMA flow chart. * did not meet inclusion criteria during title and abstract screening.

**Table 1 healthcare-13-02690-t001:** Study characteristics of included studies.

Author, Year	Country or Region of Study	Study Design	Population	Sample Size	Topic
Agroia et al., 2025 [29]	United States	Qualitative	GBMSM and transgender	47	Mpox vaccine hesitancy
Bergman et al., 2022 [30]	United States	Case study	GBMSM	1	Nursing interventions for mpox stigma prevention
Biesty et al., 2024 [31]	United Kingdom	Qualitative (interviews and workshops)	GBMSM and community leaders	Unspecified	Community-led interventions to reduce stigma and build trust
Bulcão et al., 2024 [32]	Brazil	Qualitative (online survey)	GBMSM	67	Health vulnerability in the context of the mpox epidemic
Bulcão et al., 2024 [33]	Brazil	Qualitative (online survey)	MSM	727	Self-care promotion in the context of mpox transmission
Carpino et al., 2025 [34]	United States	Cross-sectional study (online survey)	GBMSM	824	Mpox-related stigma among social networks with higher exposure to mpox
Chan et al., 2024 [35]	Singapore	Cross-sectional study (online survey)	GBMSM	237	Mpox vaccine willingness and vaccine-related communications
Chang et al., 2024 [36]	United States	Cross-sectional study (online study)	Sexual minority (SM)	79	Mpox fear and experiences of violence and discrimination
Choi et al., 2023 [37]	Republic of Korea	Randomised control trial (survey experiment)	Adults	1500	Risk perceptions, vaccine acceptance, and support for stigmatising policies through different communications strategies
Dos Santos et al., 2025 [38]	France	Qualitative (panel interviews)	Gay cisgender men	7	Barriers to accessing care and managing disease symptoms
Dsouza et al., 2023 [39]	Global, not specified	Qualitative (natural language processing)	LGBTQ+ community on Twitter	70,832 *	Online stigma against mpox among the LGBTQ+ community
Gairal-Casadó et al., 2023 [40]	Spain	Qualitative (social media analytics)	Online community engaged in mpox discourse	2313 *	Influences of global communication strategies on public discourse regarding mpox and MSM
Gilmore et al., 2024 [41]	Ireland	Qualitative (online survey)	GBMSM	163	Fear, othering, and mpox-related information consumption patterns
Gim, 2023 [42]	United States	Qualitative (natural language processing)	Online community engaged in mpox discourse	23,734 *	Mpox-related stigma communication online
Grant and Halaly, 2024 [43]	United States	Media content analysis	Media representations of GBMSM	Unspecified	Media representation and its role in reinforcing stigma
Harris et al., 2025 [44]	United States	Qualitative (interviews)	Black and Latino SM men (BLSMM)	41	Negative impacts of mpox-related stigma, vaccine scepticism, and poor communication and outreach efforts
Hong, 2023 [45]	Not applicable	Media content analysis	GBMSM	809 *	Anonymous discussion forum provides safe space for GBMSM to gather information without fear of exposure
Hughes et al., 2024 [46]	United States	Qualitative (interviews)	Gay men	30	Widespread mpox vaccine support and acceptability as a community health measure despite ongoing barriers to access
Jiao et al., 2023 [47]	China	Cross-sectional study (online survey)	GBMSM	2493	High perceived severity coupled with low perceived susceptibility indicates need for better, non-stigmatising risk communication
Kunnuji et al., 2024 [48]	Nigeria	Qualitative (interviews)	GBMSM/MSM	28	Sociolegal contexts influence awareness and knowledge about mpox, and experience seeking mpox-related care
Kutalek et al., 2025 [49]	Poland, Serbia, and Spain	Qualitative (interviews)	GBMSM	19	A multipronged, collaborative, and intersectoral approach to communication and mitigating stigma associated with challenging sociopolitical dynamics
Le Forestier et al., 2024 [50]	Australia, Canada, United Kingdom, and United States	Longitudinal study (survey)	SMM	685	The impacts of stigmatising health communication on well-being among higher risk groups
Le Forestier et al., 2024 [51]	Australia, Canada, United Kingdom, and United States	Longitudinal study (survey)	SMM	685	Stigma impacts utilisation of public health measures
Li et al., 2024 [52]	China	Cross-sectional survey	Young MSM (YMSM)	2493	Mental health challenges among YMSM during concurrent public health crises
Linares-Navarro et al., 2025 [53]	Spain	Cross-sectional survey	MSM with and without mpox diagnosis	115	Psychosocial impact and stigma experiences among MSM in Spain
Liu et al., 2025 [54]	China	Cross-sectional survey	MSM	356	Healthcare-seeking behaviour and stigma in Chinese MSM
May et al., 2023 [55]	United Kingdom	Qualitative interviews	GBMSM	44	Public health messaging, stigma, and behaviour change
Movahedi Nia et al., 2023 [56]	Canada, India, Ireland, Italy, Portugal, Spain, United Kingdom, and United States	Media content analysis	GBMSM	125,424 *	Mpox-related stigmatisation on popular social media platforms
Nerlich and Jaspal, 2025 [57]	Not applicable	Media content analysis	GBMSM	91 *	Social representations of mpox
Ogunbajo et al., 2023 [58]	United States	Cross-sectional survey	Black SMM (BSMM)	178	Mpox vaccine acceptability and access
Otmar and Merolla, 2025 [59]	United States	Longitudinal survey	GBMSM	254	Social determinants of health and impact of mpox on mental health
Owens and Hubach, 2023 [60]	United States	Cross-sectional survey	Sexual and gender minority (SGM) assigned male at birth (AMAB)	496	Consumption and attitudes towards mpox-related media
Quispe and Castagnetto, 2023 [61]	Latin America and the Caribbean	Mixed methods	GBMSM	Unspecified	Challenges controlling mpox transmission
Santos et al., 2024 [62]	Brazil	Cross-sectional survey	MSM	1452	Intersection of chemsex and public health
Santos et al., 2024 [63]	Brazil	Cross-sectional survey	MSM	1449	Factors associated with mpox vaccine hesitancy
Schmalzle et al., 2024 [64]	United States	Cross-sectional survey	MSM with mpox diagnosis	32	Stigma experiences among MSM diagnosed with mpox
Shen et al., 2024 [65]	Taiwan	Cross-sectional survey	MSM receiving mpox vaccination	2827	Vaccine-related stigma and preferences for injection site
Smith et al., 2024 [66]	Australia	Qualitative interviews	GBMSM	16	Experiences among people diagnosed with mpox or in close proximity to someone diagnosed with mpox
Takenaka et al., 2024 [67]	United States	Qualitative interviews	Gay, bisexual, and other sexually minoritised men (GBSMM)	33	Experiences with the mpox outbreak
Torres et al., 2023 [68]	Brazil	Cross-sectional online survey	Sexual and gender minorities (mostly GBMSM)	6236	Knowledge, stigma, and willingness to vaccinate in SGM
Turpin et al., 2023 [69]	United States	Qualitative interviews	BSMM	24	Experiences and perceptions of mpox, mpox-related stigma, and HIV pre-exposure prophylaxis (PrEP) engagement
Witzel et al., 2024 [70]	United Kingdom	Qualitative interviews	Cisgender and transgender GBMSM diagnosed with mpox and clinical/community-based stakeholders	26	Experiences associated with mpox and clinical and social support needs
Xu et al., 2024 [71]	China	Cross-sectional survey	YMSM	2493	Self-isolation and other risk mitigation behaviours among YMSM diagnosed with mpox
Yang et al., 2025 [72]	United States	Cross-sectional survey	GBMSM	439	Mpox perceptions and preventive behaviours
Zhang et al., 2024 [73]	China	Qualitative interviews	GBMSM	15	Mpox patient experiences and advice for control and prevention of spread
Zhang et al., 2025 [74]	China	Cross-sectional survey	MSM	2403	Psychosocial correlates of mpox vaccination
Zimmermann et al., 2023 [75]	Netherlands	Cross-sectional survey	MSM	394	Anticipated stigma and comparison with other infections

* indicates number of unique pieces of media (e.g., news articles) or posts on a social media platform (e.g., Twitter [X] or Reddit).

## Data Availability

No new data were created or analysed in this study. Data sharing is not applicable to this article.

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
