# Peer review of "Mpox-Related Stigma Among Gay, Bisexual, and Other Men Who Have Sex with Men: A Narrative Review"

_healthcare, 2025, doi:10.3390/healthcare13212690_

Round 1

Reviewer 1 Report

Comments and Suggestions for Authors

Thank you for the opportunity to review this very interesting and important manuscript.  Overall, the quality of the manuscript is high and there are only a few grammatical and spelling errors that will need to be corrected before potential publication.

Please find below a detailed listing of suggestions and comments for the authors:

General comment:

  1. I would suggest that the authors review the manuscript for consistency in terms of their use of "Mpox" vs. "mpox".  It is my understanding that "Mpox" is to be used only if this is at the beginning of a sentence and otherwise, "mpox" is to be used, but this is not what I see in the manuscript.

Introduction

  1. "Clade I primarily....Africa as travelled-associated infections." - I believe the authors mean to say "travel-associated infections" here
  2. "Lesbian, gay, bisexual, transgender, queer and questioning (LGBTQ+) people tend to utilise..." - The + in the LGBTQ+ acronym does not refer solely to people who are questioning.  This should be modified to reflect the broader meaning and range of possible sexual and gender identities of the + in the acronym.
  3. "This is evident even in healthcare....interpersonal barriers." - This sentence is worded in an awkward manner and requires clarification.
  4. "Healthcare professionals tend to lack....gaps evident both implicitly and explicitly." - I would recommend expanding a bit on what the authors mean by implicit and explicit gaps

Figure 1

  1. What does the **  after "records excluded" mean?  I can't see the explanation for the  **  anywhere.
  2. The reasons for excluding the 49 records is not clear in this figure.  Please clarify.

Results

Media Influence

  1. "A separate content analysis...., recycled conspiracies about punishment...." - suggest changing "conspiracies" to "conspiracy theories" here
  2. "Despite these efforts, the infodemic illustrated..." - suggest the authors define the word "infodemic"

Public Health Messaging

  1. "Researchers also highlighted...believed LGBTQIA+ [lesbian, gay, bisexual, transgender, queer, intersex, asexual, plus] people...." - The "plus" should be explained further.
  2. "Those diagnosed tended to be younger,...to change sexual behaviour...." - what do the authors mean by "change sexual behaviour"

Discussion

  1. "Similarly, healthcare professionals...contact tracing investigations." - This sentence is worded in an awkward way.  Suggest revising for clarity.
  2. "There was some evidence of community-driven engagement...underground services." - Another sentence that is worded awkwardly.  Pleased revise for clarity.
  3. "The move from Twitter to X has resulted increased hate speech including..." - suggest changing to "...resulted in increased hate speech, including..."

Conclusion

  1. "The initial focus on GBMSM as high-risk group..." - add in the word "a" before "high-risk group"

Author Response

Dear reviewer,

Thank you very much for the time and care you dedicated to reviewing our manuscript. We truly appreciate your thoughtful comments and constructive feedback. Please find the reviewer response form attached.

Yours sincerely,

Matthew Berger

Reviewer 2 Report

Comments and Suggestions for Authors

Thank you for the opportunity to review your work. My suggestions and comments are below:

1) The background of the research is sturdy but could be improved by stating clearly why GBMSM are more at risk of Mpox infection and are therefore the focus of this paper? Is this because of lack of public health education or the openness of this population to engage in sex with partners without discussing sexual histories and/or engaging in sexual activity with anonymous sex partners? Please integrate this into the paper. Thank you.

2) The methodology is well detailed. However, a recommendation is to have a search strategy table in place with all the pertinent research information contained in this table. The authors may refer to the PRISMA-ScR guidelines as per Tricco et al. (2018) for reference. Thank you. 

3) In the section "Community Responses" the first paragraph has a lot of information but its link to stigma is tenuous or nonexistent. Kindly rework this paragraph to reflect how social networks etc. are connected to stigma or work against stigma in the context of mpox. Thank you. 

Author Response

(The authors gave the same response as above.)

Reviewer 3 Report

Comments and Suggestions for Authors

Dear authors,

Thank you for the opportunity to review your manuscript. You are undertaking an important project as well as one that is interesting. Examining stigma related to mpox in healthcare contexts, particularly among men who have sex with men (MSM), is an under researched area. As you point out, mpox outbreaks have revealed both echoes of earlier epidemics such as HIV as well as unique challenges in how healthcare providers and the public frame disease and identity. Your work promises to fill a significant gap by analyzing these discourses, with implications for reducing stigma and improving health outcomes. I was excited to receive the invitation to review this paper, and I am impressed with the methodological rigor and the clarity of your thematic findings.

I would like to begin by pointing out what I see as the primary strengths of your work. As you continue to revise the paper and prepare it for publication, please consider how you can keep these elements intact:

  1. Methodological soundness: In my opinion, the depth of analysis demonstrates careful and systematic work. I also appreciate your transparency in divulging your methods.
  2. Thematic contributions: The themes make sense and are well-developed in terms of the goals of your research. I found it particularly interesting your identification of mpox stigma following similar patterns to HIV stigma, as this historical continuity is important for public health scholars and practitioners.
  3. Clarity of implications: You connect your findings clearly to practice and policy, providing insights that can shape both healthcare provider education and stigma-reduction efforts. Well done here!

Although I am a fan of what you have done so far, I have a few suggestions that I believe will strengthen the paper further:

  • Language use: Please be cautious with the term homosexual unless you are directly quoting from sources or providing historical context. In many scholarly and activist communities, the term is considered outdated and pathologizing. Alternatives such as “gay men,” “queer men,” or “men who have sex with men (MSM)” are more appropriate depending on the context.
  • Minor editorial fixes: There are a few places where small editing for flow, transitions, or clarity would help readability. These are minor, but tightening prose will help your arguments stand out more strongly.
  • Nuancing comparisons: While I found the HIV–mpox parallels fascinating, you might consider briefly noting the limits of comparison as well, That ensures readers understand that while patterns repeat, contexts also shift.

As you will note, all of my concerns are relatively minor and readily fixable.

Overall, my opinion is that this is a well-prepared and insightful manuscript. With some small refinements, it will be in great shape.

Thank you again for the opportunity to review this work. I am excited to see the final published version and the contributions it will make to scholarship on stigma, sexuality, and health communication.

Author Response

(The authors gave the same response as above.)
